# Analysis of Controlled Rabi Flopping in a Double Rephasing Photon Echo Scheme for Quantum Memories

**DOI:** 10.3390/e22091007

**Published:** 2020-09-09

**Authors:** Rahmat Ullah, Byoung S. Ham

**Affiliations:** 1Center for Photon Information Processing, and School of Electrical Engineering and Computer Science, Gwangju Institute of Science and Technology, Gwangju 61005, Korea; rahmatktk@comsats.edu.pk; 2Quantum Optics Lab., Department of Physics, COMSATS University, Islamabad 44000, Pakistan

**Keywords:** quantum optics, coherent transients, quantum memory, photon echoes

## Abstract

A double rephasing scheme of a photon echo is analyzed for inversion-free photon echo-based quantum memories using controlled Rabi flopping, where the Rabi flopping is used for phase control of collective atom coherence. Unlike the rephasing-caused π-phase shift in a single rephasing scheme, the control Rabi flopping between the excited state and an auxiliary third state induces coherence inversion. Thus, the absorptive photon echo in a double rephasing scheme can be manipulated to be emissive. Here, we present a quantum coherence control of atom phases in a double rephasing photon echo scheme for emissive photon echoes for quantum memory applications.

## 1. Introduction

Modified photon echoes have been intensively studied for quantum memory applications over the last decade since the first protocol of controlled reversible inhomogeneous broadening, where an efficient photon echo can be achieved by a counter-propagating control pulse set in a three-level Doppler [1] and non-Doppler medium [2]. Due to the inherent population inversion in photon echoes [3], resulting in quantum noises and violation of the no cloning theorem [4], a conventional photon echo itself cannot be directly applied to quantum memories. Compared with single-atom-based quantum memory protocols, e.g., utilizing nuclear spins recently demonstrated in Si-based semiconductors [5], the photon echoes in rare-earth doped solids have benefits of multimode, ultrafast, and ultrahigh absorption [6]. To overcome the inherent population inversion in photon echoes [3], atomic frequency comb (AFC) echoes [7,8], gradient echoes [9,10], and controlled double rephasing (CDR) echoes [11,12,13] are presented for quantum memory applications. Because ultralong quantum memory is an essential condition for long-distance quantum communications using quantum repeaters [14,15], storage time extension has also been a critical issue [16,17,18,19]. As experimentally demonstrated by using dynamic decoupling (DD) [16] and optical locking via controlled coherence conversion (CCC) [20], the optical storage time can be extended up to spin population decay time.

The CCC theory was proposed to convert the absorptive echo into an emissive one in a double rephasing (DR) photon echo scheme [11]. The DR photon echo scheme inherently gives the benefit of no population inversion. Because a π-rephasing pulse induces reversible coherence evolutions in a time domain with a π-phase shift, the DR echo is obviously absorptive like the data pulse due to the 2π-phase shift (no change) in the collective coherence. Regardless of silent echoes in the DR scheme [21,22,23], the collective coherence of the final echo is the sum of individual coherence evolutions, resulting in absorptive coherence [11,12]. Moreover, there is no way to solve this absorptive echo problem in a two-level system, at least not yet. It should be noted that the photon echo in an ensemble system must be distinguished from a single atomic system. Although population inversion or the sign of coherence has nothing to do with the single qubit system, it is critical to the ensemble system because of the macroscopic coherent transients of a nonlinear effect. The π–π control pulse-induced negative sign of ensemble coherence can never be radiated out of the medium regardless of population inversion [11,17,23].

DR photon echoes, however, have been observed, which is seemingly violating the CDR echo theory [20,21,22]. The reason of seemingly contradiction in [20,21,22] is due to the imperfect rephrasing pulse area caused by a Gaussian rephrasing pulse. Recently, such Gaussian pulse-caused echo generation was discussed to be as high as 26% in retrieval efficiency [24]. The CCC in CDR echoes was already discussed in a single rephasing photon echo scheme theoretically [25,26], as well as experimentally [27]. Here, in the present paper, we analytically investigate the collective atom phase shift in the DR scheme and confirm the CDR echo theory with proof of coherence inversion. Compared with full numerical analysis in previous discussions [11,12,17,18,25], we present full analytic solutions in this article. 

## 2. Theory 

Figure 1 is a schematic diagram of the present CDR echoes, where the control pulse set of C_1_ and C_2_ is for the atom phase control in the DR scheme. The data (D), first rephasing (R_1_), and second rephasing (R_2_) pulses satisfy a DR photon echo scheme, where they are resonant between states |1〉 and |2〉 as shown in Figure 1a. The pulse sequence of CDR is shown in Figure 1b, where the control pulse set C_1_ and C_2_ is resonant between states |2〉 and |3〉. The time delay τ between C_1_ and C_2_ is used for storage time extension, which is limited by the spin dephasing [25,26,27]. The spin dephasing can be minimized with the zero first-order Zeeman method [28]. In an optical locking scheme applied to three-pulse photon echoes [18], the storage time extends up to spin population decay time [20]. To satisfy general conditions of CDR, each pulse area of R_1_, R_2_, C_1_, and C_2_ is set to be π. The pulse area of D is set to small at 0.1π. The pulse area is defined by $\varphi_{i}=\int \Omega_{i}\mathrm{dt},$ and $\Omega_{i}\;$(i= D, R_1_, R_2_, C_1_, and C_2_) is the Rabi frequency of the pulse.

The CCC in CDR echoes must be distinguished from resonant Raman or electromagnetically induced transparency (EIT) based on two-photon resonance without shelving on the excited state. For photon echo-based quantum memories, the signal (data) pulse information (phase and amplitude) must be fully transferred into a matter (optical coherence) state via a complete absorption process in an optically dense, inhomogeneously broadened two-level medium [17]. Unlike other coherence optics in the three-level system mentioned above, the inhomogeneity of the ensemble is the fundamental requirement for the coherence evolutions in photon echoes. One unique property of the CCC is the double coherence swapping between the optical and spin states via the control pulse set of C_1_ and C_2_. Unlike EIT, the R_1_ and C_1_ must be differentiated from the two-photon Raman coherence, where the delay ΔT between R_1_ and C_1_ must be longer than the inverse of inhomogeneous width. Usually, this requirement is easily satisfied for the consecutive π-optical pulse sequence in most rare-earth doped solids [27].

The purpose of C_1_ is simply to hold both optical phase decay via complete population transfer from the excited state |2〉 to the auxiliary spin state |3〉, resulting in optical–spin coherence conversion with *ρ*_12_ = 0 [11]. For this, the state |3〉 must be set to be vacant initially. When the second control pulse C_2_ is turned on, the system population is completely recovered to the initial one reached by R_1_. However, the system coherence is not invariant due to the π-phase shift, resulting in absorptive photon echo E_2_ [11,12,13,18,24,25,26,27].

The interaction picture Hamiltonian in the atom–field interactions under rotating-wave approximation of the proposed system in Figure 1a is written as
(1)$$H=-\hbar /2\left[\begin{array}{ccc} 0 & \Omega_{j} & 0\\ \Omega_{j} & 0 & \Omega_{k}\\ 0 & \Omega_{k} & 0 \end{array}\right]$$
where $\Omega_{j}\;$(j = D, R_1_, or R_2_) is the Rabi frequency of D, R_1_, and R_2_, and $\Omega_{k}$ (k = C_1_ or C_2_) is the Rabi frequency of C_1_ or C_2_. We calculate the rate equations for the density matrix elements using the Von Neumann equation [29].
(2)$$\dot{\rho }=-\frac{i}{\hbar }\left[H,\rho \right]-\frac{1}{2}\left\{\Gamma ,\rho \right\}$$

The corresponding rate equations are
(3)$${\dot{\rho }}_{11}=-i\frac{\Omega_{j}}{2}\left(\rho_{12}-\rho_{21}\right),$$
(4)$${\dot{\rho }}_{22}=-i\frac{\Omega_{j}}{2}\left(\rho_{21}-\rho_{12}\right)-i\frac{\Omega_{k}}{2}\left(\rho_{23}-\rho_{32}\right),$$
(5)$${\dot{\rho }}_{33}=-i\frac{\Omega_{k}}{2}\left(\rho_{32}-\rho_{23}\right),$$
(6)$${\dot{\rho }}_{12}=-i\frac{\Omega_{j}}{2}\left(\rho_{11}-\rho_{22}\right)-i\frac{\Omega_{k}}{2}\rho_{13},$$
(7)$${\dot{\rho }}_{12}=-i\frac{\Omega_{k}}{2}\rho_{12}+i\frac{\Omega_{j}}{2}\rho_{23},$$
(8)$${\dot{\rho }}_{23}=-i\frac{\Omega_{k}}{2}\left(\rho_{22}-\rho_{33}\right)+i\frac{\Omega_{j}}{2}\rho_{13},$$
where all decay rates are set to zero for simplicity. We now consider the CDR echo scheme for the discussion below. For this, we start with a general DR scheme to investigate the absorptive coherence of the final echo E_2_ without C_1_ and C_2_ pulses in Figure 1.

## 3. Discussion

### 3.1. DR Photon Echoes

In this subsection, we study conventional two-pulse photon echoes in a DR scheme without C_1_ and C_2_ in Figure 1. We derive time-dependent density matrix equations for the expressions of coherence between the ground and excited states and the population in each bare state.

#### 3.1.1. D-Pulse

We first derive the expressions of coherence and population excited by the D-pulse. The equations of motion for D-pulse by setting $\Omega_{j}=\Omega_{D}$ and $\Omega_{k}=0$ in Equations (3)–(8) are as follows:(9)$${\dot{\rho }}_{11}=-i\frac{\Omega_{D}}{2}\left(\rho_{12}-\rho_{21}\right),$$
(10)$${\dot{\rho }}_{22}=-i\frac{\Omega_{D}}{2}\left(\rho_{21}-\rho_{12}\right),$$
(11)$${\dot{\rho }}_{12}=-i\frac{\Omega_{D}}{2}\left(\rho_{11}-\rho_{22}\right),$$
(12)$${\dot{\rho }}_{21}=-i\frac{\Omega_{D}}{2}\left(\rho_{22}-\rho_{11}\right).$$

Initially, all atoms are in the ground state |1〉: $\rho_{11}\left(0\right)=1;$
$\rho_{22}\left(0\right)=\rho_{12}\left(0\right)=\rho_{21}\left(0\right)=0.$ The Laplace transform of Equations (9)–(12) with $\rho_{11}+\rho_{22}=1$ yields
(13)$$ℒ\left[\rho_{11}\right]=\frac{2s^{2}+\Omega_{D}^{2}}{2s\left(s^{2}+\Omega_{D}^{2}\right)},$$
(14)$$ℒ\left[\rho_{12}\right]=\frac{-i\Omega_{D}}{\left(s^{2}+\Omega_{D}^{2}\right)},$$
(15)$$ℒ\left[\rho_{21}\right]=\frac{i\Omega_{D}}{\left(s^{2}+\Omega_{D}^{2}\right)}.$$

The final equations for population and coherence are obtained by taking the inverse Laplace transform of Equations (13)–(15).
(16)$$\rho_{11}=\cos^{2}\left(\frac{\varphi_{D}}{2}\right),$$
(17)$$\rho_{22}=\sin^{2}\left(\frac{\varphi_{D}}{2}\right),$$
(18)$$\rho_{12}=-\frac{i}{2}\sin\left(\varphi_{D}\right),$$
where $\varphi_{D}$ is the area of the D-pulse. The D-pulse obeys the area theorem which has a direct relationship with coherence [30].
(19)$$\frac{\partial \varphi_{D}}{\partial z}=-\frac{\alpha }{2}\sin\left(\varphi_{D}\right),$$
where α is the absorption coefficient. For the D-pulse having a very small area, $\sin\left(\varphi_{D}\right)\approx 1$, $\varphi_{D}={\left(\varphi_{D}\right)}_{0}e^{-\alpha z/2}$ representing Beer′s law. The information of D-pulse is now transferred into the ensemble coherence. For a weak D-pulse $\varphi_{D}\ll 1$, the atomic population still remains in the ground state |1〉: $\rho_{11}\approx 1$; $\rho_{22}\approx 0$. In our analysis, the D-pulse area is set to be 0.1π. 

#### 3.1.2. R_1_-Pulse

As soon as the atoms are excited by D, they immediately start to evolve with their own detuning-dependent phase velocity until the rephasing pulse (R_1_-pulse) comes. We use Equations (16)–(18) as initial conditions for $\Omega_{j}=\Omega_{R_{1}}$ and $\Omega_{k}=0$ to solve Equations (3)–(8). The solution of the rate equations for R_1_-pulse is as follows:(20)$$\rho_{11}=\cos^{2}\left(\frac{\varphi_{D}+\varphi_{R_{1}}}{2}\right),$$
(21)$$\rho_{22}=\sin^{2}\left(\frac{\varphi_{D}+\varphi_{R_{1}}}{2}\right),$$
(22)$$\rho_{12}=-\frac{i}{2}\sin\left(\varphi_{D}+\varphi_{R_{1}}\right),$$
where $\varphi_{R_{1}}$ is the pulse area of R_1_. Equation (22) indicates that the rephasing π pulse R_1_ results in a π shift in the coherence $\rho_{12}$ initiated by the D-pulse in Equation (18) as shown in Figure 2a: $\left[\rho_{12}\right]\overset{R1}{\to }{\left[\rho_{12}\right]}^{*}$ (see also Appendix A for π/2 pulse area of D). All details of detuning-dependent atom phase evolutions and rephasing effects are numerically shown in Figure 4 of [25], where the real parts of $\rho_{12}$ are exactly symmetric, cancelling each other’s coherence. The π rephasing pulse swaps the population between ground and excited states as shown in Figure 2b, resulting in spontaneous and/or stimulated emission. To overcome the population inversion, a controlled double rephasing concept was developed in the name of CDR echoes [11,12]. For DR echoes, the second π optical pulse R_2_ is added to swap the populations again, where the second echo E_2_ is free from quantum noises. To fully restore the D-pulse transferred coherence, the first echo E_1_ must be erased (or silenced), where the silent echo does not affect the individual coherence evolutions [21,22,23]. 

We derive the coherence and population rate equations for R_2_-pulse by replacing $\Omega_{j}$ with $\Omega_{R_{2}}$ and setting $\Omega_{k}=0$ in Equations (3)–(8). We use Equations (20)–(22) as the initial conditions and calculate the expressions for coherence and populations as follows:(23)$$\rho_{11}=\cos^{2}\left(\frac{\varphi_{D}+\varphi_{R_{1}}+\varphi_{R_{2}}}{2}\right),$$
(24)$$\rho_{22}=\sin^{2}\left(\frac{\varphi_{D}+\varphi_{R_{1}}+\varphi_{R_{2}}}{2}\right),$$
(25)$$\rho_{12}=-\frac{i}{2}\sin\left(\varphi_{D}+\varphi_{R_{1}}+\varphi_{R_{2}}\right).$$

In Figure 2c,d, the R_2_ pulse area-dependent coherence and population are shown for $\varphi_{D}=0.1\pi$ and $\varphi_{R_{1}}=\pi$. As shown in Figure 2c, the π-R_2_ pulse inverts the coherence as the π-R_1_ pulse does. Here, the negative sign in the coherence $\rho_{12}$ shows absorption. Thus, the second echo by R_2_ is absorptive like the data pulse D [11,12]. This means that the generated echo E_2_ in the DR scheme cannot be radiated out of the medium due to the coherent transient effects, as D is fully absorbed into the medium. By the way, the observations of E_2_ in [21,22,23] were understood as imperfect rephasing-caused coherence leakage due to Gaussian distributed light pulses [24]. Our aim here is to get the inversion-free emissive echo. To convert the absorptive echo E_2_ in Figure 2 into an emissive one, the CDR echo scheme is applied. In the section below, we describe the roles of C_1_ and C_2_ for CCC in detail.

### 3.2. CDR Photon Echoes

In this subsection, we discuss the CDR echo of Figure 1 by inserting the control pulse set of C_1_ and C_2_ in the DR scheme. The control pulse set position can be after either R_1_ as shown in Figure 1b or R_2_ as discussed in [11,12]. In both cases, C_1_ must be activated before the echo timing [25].

#### 3.2.1. C_1_-Pulse

The function of C_1_-pulse with a π-pulse area is to temporally hold optical coherence decay, as well as optical phase evolution, via transferring population in the excited state |2〉 to an auxiliary ground (spin) state |3〉. In general, spin phase decay rate is much longer than the optical counterpart in rare-earth doped crystals. Thus, C_1_ plays the role of storage time extension [12,18]. The coherence and population changes by C_1_ can be obtained using Equations (20)–(22) as initial conditions. The solutions of density matrix Equations (3)–(8) for C_1_ are obtained as
(26)$$\rho_{11}=\cos^{2}\left(\frac{\varphi_{D}+\varphi_{R_{1}}}{2}\right),$$
(27)$$\rho_{22}=\cos^{2}\left(\frac{\varphi_{C_{1}}}{2}\right)\sin^{2}\left(\frac{\varphi_{D}+\varphi_{R_{1}}}{2}\right),$$
(28)$$\rho_{33}=\sin^{2}\left(\frac{\varphi_{C_{1}}}{2}\right)\sin^{2}\left(\frac{\varphi_{D}+\varphi_{R_{1}}}{2}\right),$$
(29)$$\rho_{12}=-\frac{i}{2}\cos\left(\frac{\varphi_{C_{1}}}{2}\right)\sin\left(\varphi_{D}+\varphi_{R_{1}}\right),$$
(30)$$\rho_{13}=-\frac{1}{2}\sin\left(\frac{\varphi_{C_{1}}}{2}\right)\sin\left(\varphi_{D}+\varphi_{R_{1}}\right),$$
(31)$$\rho_{23}=-\frac{i}{2}\sin\left(\varphi_{C_{1}}\right)\sin^{2}\left(\frac{\varphi_{D}+\varphi_{R_{1}}}{2}\right).$$

The optical coherence $\rho_{12}$ in Equation (29) by C_1_-pulse is equal to $\cos\left(\varphi_{C_{1}}/2\right)$ times the coherence generated by R_1_-pulse in Equation (22), where the R_1_-resulted coherence is 0.15 for the 0.1π of D-pulse and π of R1-pulse (see Figure 3). Thus, Equation (29) becomes $\rho_{12}=0.15i\cos\left(\varphi_{C_{1}}/2\right)$ (see also Figure 2a). Similarly, the spin coherence in Equation (30) is $\rho_{13}=0.15\sin\left(\varphi_{C_{1}}/2\right).$ In the absence of the C_1_-pulse, i.e., $\varphi_{C_{1}}=0,$
$\rho_{12}=0.15i$ and $\rho_{13}=0$. In the presence of the π C_1_-pulse, the optical and spin coherence becomes $\rho_{12}=0.15i\cos\left(\pi /2\right)=0$ and $\rho_{13}=0.15\sin\left(\pi /2\right)=0.15ie^{-i\pi /2}$, respectively. The π-C_1_-pulse adds a π/2 phase shift to the transferred coherence $\rho_{13}$ [28]. This is a well-known property in resonant two-field interactions in a three-atomic system, where there is a π/2 phase shift between *Im*$\left[\rho_{12}\right]$ and *Re*$\left[\rho_{13}\right]$ In conclusion, the C_1_-pulse locks both optical phase decay and coherence evolutions, while it transfers $\rho_{12}$ into $\rho_{13}$ with a π/2 phase shift via complete population transfer. In other words, *Im*$\left[\rho_{12}\right]$ becomes *Re*$\left[\rho_{13}\right]$ as shown in Figure 3a. Here, *Im*$\left[\rho_{13}\right]$ is zero as *Re*$\left[\rho_{12}\right]$ is zero in Equations (16)–(18). 

#### 3.2.2. C_2_-Pulse

The function of C_2_-pulse is to restore the transferred coherence by C_1_. Using Equations (26)–(31) as the initial conditions, and setting $\Omega_{k}\;=\Omega_{C_{2}}$ and $\Omega_{j}\;=0$ in Equations (3)–(8), the system coherence and population expressions by C_2_-pulse are obtained as follows:(32)$$\rho_{11}=\cos^{2}\left(\frac{\varphi_{D}+\varphi_{R_{1}}}{2}\right),$$
(33)$$\rho_{22}=\cos^{2}\left(\frac{\varphi_{C_{1}}+\varphi_{C_{2}}}{2}\right)\sin^{2}\left(\frac{\varphi_{D}+\varphi_{R_{1}}}{2}\right),$$
(34)$$\rho_{33}=\sin^{2}\left(\frac{\varphi_{C_{1}}+\varphi_{C_{2}}}{2}\right)\sin^{2}\left(\frac{\varphi_{D}+\varphi_{R_{1}}}{2}\right),\;$$
(35)$$\rho_{12}=-\frac{i}{2}\cos\left(\frac{\varphi_{C_{1}}+\varphi_{C_{2}}}{2}\right)\sin\left(\varphi_{D}+\varphi_{R_{1}}\right),$$
(36)$$\rho_{13}=-\frac{1}{2}\sin\left(\frac{\varphi_{C_{1}}+\varphi_{C_{2}}}{2}\right)\sin\left(\varphi_{D}+\varphi_{R_{1}}\right),$$
(37)$$\rho_{23}=-\frac{i}{2}\sin\left(\varphi_{C_{1}}+\varphi_{C_{2}}\right)\sin^{2}\left(\frac{\varphi_{D}+\varphi_{R_{1}}}{2}\right).$$

The coherence in Equation (35) is equal to $\cos\left(\left(\varphi_{C_{1}}+\varphi_{C_{2}}\right)/2\right)$ multiplied by the coherence excited by R_1_-pulse in Equation (22). The π–π pulse sequence of C_1_ and C_2_, therefore, induces a coherence inversion via the round trip of population transfer between the excited and auxiliary states: cos((π+π)/2)=−1 (see Figure 3c) [11,12,24]: $\rho_{12}\overset{C_{1}\&C_{2}}{\to }-\rho_{12}$. This coherence inversion mechanism is completely different from the rephasing by R_1_ or R_2_ [12]. In order to resume the coherence initiated by the R_1_-pulse, the sum pulse area of C_1_ and C_2_ must be equal to 4*nπ* (*n* = 1,2,3…). In Figure 3c,d, we plot the coherence and population as a function of the C_2_-pulse area for $\varphi_{D}=0.1\pi$, $\varphi_{R_{1}}=\pi$, and $\varphi_{C_{1}}=\pi$. Figure 3c shows that the ensemble coherence excited by D and rephased by R_1_ is recovered with the 3*π* C_2_-pulse. The 3π C_2_, of course, returns the population from state |3〉 to the excited state |2〉 as shown in Figure 3d. Thus, the (π–π) C_1_–C_2_ pulse sequence in a controlled AFC [31] induces an absorptive echo as in the DR scheme in Figure 2c due to the π-phase shift by the control Rabi flopping. The experimental observation in [31] is not an artefact but due to the coherence leakage through imperfect rephasing by commercial Gaussian distributed laser pulses, where its maximum echo efficiency is far less then unity [24]. In the CDR echo scheme, however, the π–π pulse sequence of C1 and C2 is required to compensate for the π-phase shift in the DR scheme.

#### 3.2.3. R_2_-Pulse

For the CDR echo in Figure 1, the final analytic solutions of density matrix Equations (3)–(8) are obtained using Equations (32)–(37) as the initial conditions.


(38)$$\begin{array}{c} \rho_{11}=\frac{1}{16}[\cos\left(\frac{\varphi_{C_{1}}+\varphi_{C_{2}}-\varphi_{D}-\varphi_{R_{2}}-\varphi_{R_{1}}}{2}\right)-\cos\left(\frac{\varphi_{C_{1}}+\varphi_{C_{2}}-\varphi_{D}+\varphi_{R_{2}}-\varphi_{R_{1}}}{2}\right)\\ +2\cos\left(\frac{\varphi_{D}-\varphi_{R_{2}}+\varphi_{R_{1}}}{2}\right)-\cos\left(\frac{\varphi_{C_{1}}+\varphi_{C_{2}}+\varphi_{D}-\varphi_{R_{2}}+\varphi_{R_{1}}}{2}\right)\\ +\cos\left(\frac{\varphi_{C_{1}}+\varphi_{C_{2}}+\varphi_{D}+\varphi_{R_{2}}-\varphi_{R_{1}}}{2}\right)+2\cos\left(\frac{\varphi_{D}+\varphi_{R_{2}}+\varphi_{R_{1}}}{2}\right){]}^{2}, \end{array}$$
(39)$$\begin{array}{c} \rho_{22}=\frac{1}{16}[\sin\left(\frac{\varphi_{C_{1}}+\varphi_{C_{2}}-\varphi_{D}-\varphi_{R_{2}}-\varphi_{R_{1}}}{2}\right)+\sin\left(\frac{\varphi_{C_{1}}+\varphi_{C_{2}}-\varphi_{D}+\varphi_{R_{2}}-\varphi_{R_{1}}}{2}\right)\\ +2\sin\left(\frac{\varphi_{D}-\varphi_{R_{2}}+\varphi_{R_{1}}}{2}\right)-\sin\left(\frac{\varphi_{C_{1}}+\varphi_{C_{2}}+\varphi_{D}-\varphi_{R_{2}}+\varphi_{R_{1}}}{2}\right)\\ -\sin\left(\frac{\varphi_{C_{1}}+\varphi_{C_{2}}+\varphi_{D}+\varphi_{R_{2}}-\varphi_{R_{1}}}{2}\right)-2\sin\left(\frac{\varphi_{D}+\varphi_{R_{2}}+\varphi_{R_{1}}}{2}\right){]}^{2}, \end{array}$$
(40)$$\rho_{33}=\sin^{2}\left(\frac{\varphi_{C_{1}}+\varphi_{C_{2}}}{2}\right)\sin^{2}\left(\frac{\varphi_{D}+\varphi_{R_{1}}}{2}\right),\;$$
(41)$$\begin{array}{cc} \rho_{12}=-\frac{i}{16}[2\sin & \left(\varphi_{R_{2}}\right)\\ & +2\sin\left(\varphi_{R_{2}}\right)\cos\left(\varphi_{D}+\varphi_{R_{1}}\right)\left(3+\cos\left(\frac{\varphi_{C_{1}}+\varphi_{C_{2}}}{2}\right)\right)\\ & +\sin\left(\varphi_{C_{1}}+\varphi_{C_{2}}-\varphi_{R_{2}}\right)\\ & -\sin\left(\varphi_{C_{1}}+\varphi_{C_{2}}+\varphi_{R_{2}}\right)\\ & +8\cos\left(\varphi_{R_{2}}\right)\sin\left(\varphi_{D}+\varphi_{R_{1}}\right)\cos\left(\frac{\varphi_{C_{1}}+\varphi_{C_{2}}}{2}\right)]. \end{array}$$


In Figure 4, we plot the evolutions of coherence and population as a function of R_2_-pulse area for $\varphi_{D}=0.1\pi ,$
$\varphi_{R_{1}}=\pi$, $\varphi_{C_{1}}=\pi$, and $\varphi_{C_{2}}=\pi$. As a result, both coherence and population excited by D are recovered with a π-pulse area of R_2_, where spontaneous or stimulated emission-caused quantum noises are completely eliminated.

The present scheme can experimentally be realized in a rare-earth Pr^3+^-doped Y_2_SiO_5_. In most rare-earth doped media, the ground state hyperfine splitting is a few tens of megahertz. Thus, tens of GHz in an optical inhomogeneous width can be sliced for multiple spectral channels for multimode quantum memory applications, where the practical parameter of optical Rabi frequency is ~MHz. For an extended storage time by C_1_, Zeeman states may be used [32].

In an atomic ensemble such as Rb vapors, Zeeman splitting may also be used, where optical polarization control is adapted to form a three-level system. However, such an atomic medium may not a good candidate for the photon-echo-based quantum memory applications simply due to fast atomic diffusion. Moreover, providing a π optical pulse in a few-ns pulse duration within the optical phase decay time is very challenging with a commercial continuous wave (CW) laser system.

## 4. Conclusions

In conclusion, we analytically presented the CDR echo protocol for spontaneous emission-free-photon echo-based quantum memory applications by combining double rephasing photon echoes with control Rabi flopping. For this, time-dependent density matrix equations were analytically solved for coherence/population evolutions to investigate the phase shift of a resonant atom. To overcome the absorptive echo problem in a bare double rephasing photon echo scheme, a consecutive π–π control pulse sequence is inserted right after the first rephasing pulse. The control pulse-generated π-phase shift was exactly compensated for with another π-phase shift resulting from the double rephasing scheme. As a result, emissive photon echoes were obtained under no population inversion.

## Figures and Tables

**Figure 1 entropy-22-01007-f001:**
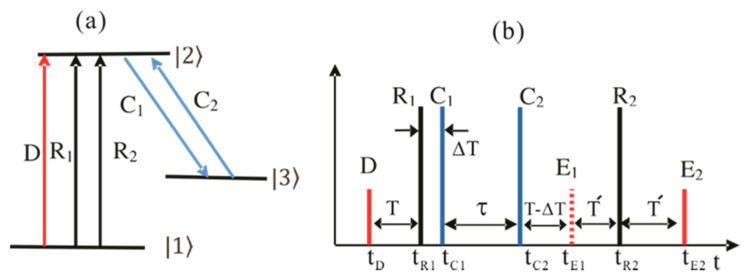
(**a**) Schematics of controlled double rephasing echoes. (**b**) Pulse sequence for (**a**), where t_j_ is the arrival time of pulse j.

**Figure 2 entropy-22-01007-f002:**
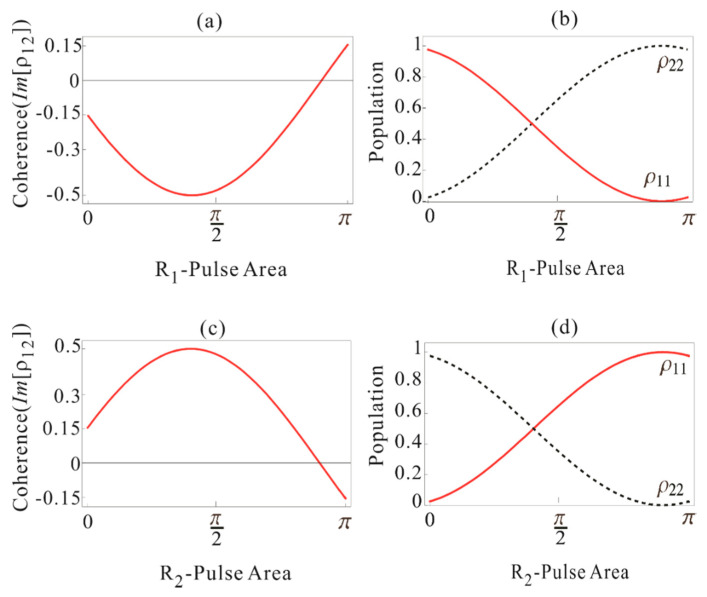
(**a**) Plot of $Im\left[\rho_{12}\right]\;$(Equation (22)) versus R_1_-pulse area $\varphi_{R_{1}}$ with area of D-pulse $\varphi_{D}$ = 0.1π. (**b**) Corresponding population evolution (red) $\rho_{11}$ (Equation (20)) and (dotted) $\rho_{22}$ (Equation (21)). (**c**) Plot of $Im\left[\rho_{12}\right]$ (Equation (25)) versus R_2_-pulse area $\varphi_{R_{2}}$ with area of D-pulse $\varphi_{D}$ = 0.1π and that of R_1_
$\varphi_{R_{1}}$ = π. (**d**) Corresponding population evolution (red) $\rho_{11}$ (Equation (23)) and (dotted) $\rho_{22}$ (Equation (24)).

**Figure 3 entropy-22-01007-f003:**
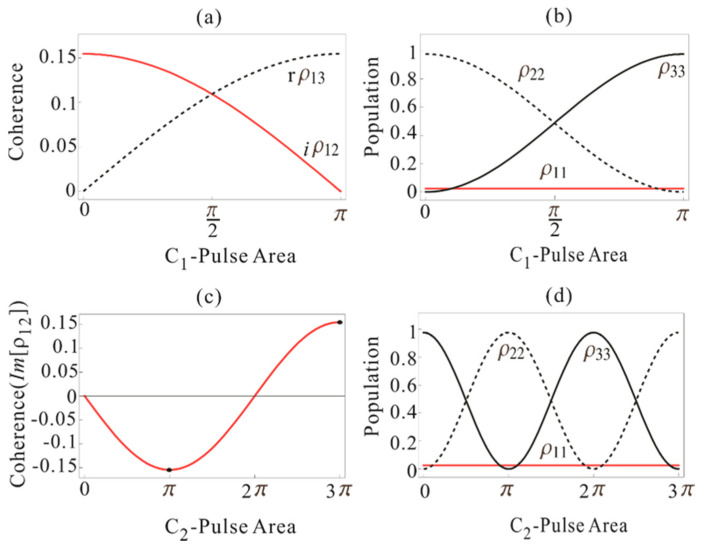
(**a**) Plot of $Im\left[\rho_{12}\right]$ and $Re\left[\rho_{13}\right]$ (Equations (29) and (30)) versus C_1_-pulse area $\varphi_{C_{1}}$ with area of D-pulse $\varphi_{D}=0.1\pi$ and that of R_1_
$\varphi_{R_{1}}=\pi$. (**b**) Corresponding population evolution (red) $\rho_{11}$ (Equation (26)), (dotted) $\rho_{22}$ (Equation (27)), and black $\rho_{33}$ (Equation (28)). (**c**) Plot of $Im\left[\rho_{12}\right]$ (Equation (35)) versus C_2_-pulse area $\varphi_{C_{2}}$ with area of other pulses $\varphi_{D}=0.1\pi$,$\;\varphi_{C_{1}}=\pi$, and $\varphi_{R_{1}}=\pi$. (**d**) Corresponding population evolution (red) $\rho_{11}$ (Equation (32)), (dotted) $\rho_{22}$ (Equation (33)), and (black) $\rho_{33}$ (Equation (34)).

**Figure 4 entropy-22-01007-f004:**
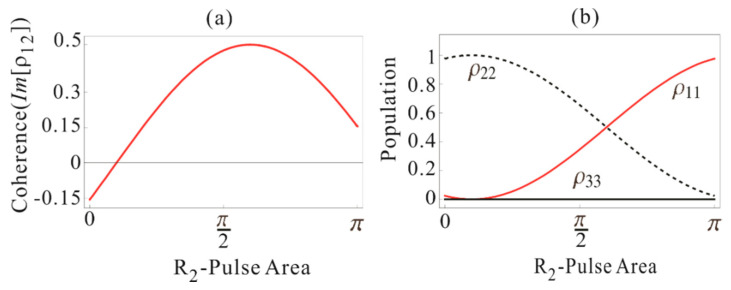
(**a**) Plot of $Im\left[\rho_{12}\right]$ (Equation (41)) versus R_2_-pulse $\varphi_{R_{2}}.$ The area of other pulses are $\varphi_{D}=0.1\pi$, $\varphi_{R_{1}}=\pi ,\;\varphi_{C_{1}}=\pi$, and $\varphi_{C_{2}}=\pi$. (**b**) Corresponding population evolution (red) $\rho_{11}$ (Equation (38)), (dotted) $\rho_{22}$ (Equation (39)), and (black) $\rho_{33}$ (Equation (40)).

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
