# Peer review of "Analysis of Controlled Rabi Flopping in a Double Rephasing Photon Echo Scheme for Quantum Memories"

_entropy, 2020, doi:10.3390/e22091007_

Round 1
Reviewer 1 Report
The manuscript (MS) analyzed a double rephasing scheme of photon echo for inversion-free photon echo based quantum memories. The MS showed the CDR echo protocol by combining double rephasing photon echoes with control Rabi flopping. Here, the Rabi flopping is used for phase control of collective atom coherence. To overcome the absorptive echo problem, consecutive Pi-Pi control pulse sequence is inserted right after the first rephasing pulse. Thus, the absorptive photon echo in a double rephasing scheme can be manipulated to be emissive. Emissive photon echoes were obtained under no population inversion.
To investigate the phase shift of a resonant atom, the MS showed detailed analytic calculations of the time-dependent density matrix equations for coherence and population of atomic levels. The MS shows how the absorptive photon echo in a double rephasing scheme can be manipulated to be emissive. Therefore, the MS is suitable to be published.
Reviewer 2 Report
The authors, in the paper entitled “Analysis of controlled Rabi flopping in a double rephasing photon echo scheme for quantum memories”, propose and analyze a double rephasing scheme of photon echo for inversion-free photon echo-based quantum memories, using controlled Rabi flopping, where the Rabi flopping is used for phase control of collective atom coherence.
The authors, in the proposed scheme, overcome the absorptive echo problem in a bare double rephasing photon echo scheme, by introducing a consecutive π-π control pulse sequence after the first rephasing pulse. In this way emissive photon echoes is obtained under no population inversion.
The introduction provide sufficient background and include all relevant references. The results are clearly presented: the time-dependent density matrix equations were analytically solved to investigate the process.
Moreover,the authors show that the scheme could be realized by exploiting rare-earth for quantum memory applications.
The paper can be published in the current version.